# The Potential Roles of Dec1 and Dec2 in Periodontal Inflammation

**DOI:** 10.3390/ijms221910349

**Published:** 2021-09-26

**Authors:** Xingzhi Wang, Fuyuki Sato, Keiji Tanimoto, Niveda Rajeshwaran, Lakshmi Thangavelu, Makoto Makishima, Ujjal K. Bhawal

**Affiliations:** 1Department of Biochemistry, Nihon University School of Medicine, Tokyo 173-8610, Japan; wangxingzhi1234@outlook.com; 2Pathology Division, Shizuoka Cancer Center, Shizuoka 411-8777, Japan; fsatodec1dec2@yahoo.co.jp; 3Department of Translational Cancer Research, Research Institute for Radiation Biology and Medicine, Hiroshima University, Hiroshima 739-8511, Japan; ktanimo@hiroshima-u.ac.jp; 4Department of Periodontics, Saveetha Dental College, Saveetha Institute of Medical and Technical Sciences, Chennai 600077, India; Nivedarajeshwaran@gmail.com; 5Department of Pharmacology, Saveetha Dental College, Saveetha Institute of Medical and Technical Sciences, Chennai 600077, India; lakshmi@saveetha.com; 6Department of Biochemistry and Molecular Biology, Nihon University School of Dentistry at Matsudo, Chiba 271-8587, Japan

**Keywords:** Dec1, Dec2, periodontal inflammation, autophagy, pyroptosis

## Abstract

Periodontal inflammation is a common inflammatory disease associated with chronic inflammation that can ultimately lead to alveolar attachment loss and bone destruction. Understanding autophagy and pyroptosis has suggested their significant roles in inflammation. In recent years, studies of differentiated embryo-chondrocyte expressed genes 1 and 2 (Dec1 and Dec2) have shown that they play important functions in autophagy and in pyroptosis, which contribute to the onset of periodontal inflammation. In this review, we summarize recent studies on the roles of clock genes, including Dec1 and Dec2, that are related to periodontal inflammation and other diseases.

## 1. Introduction

Periodontal inflammation is a prevalent oral disease characterized by chronic inflammation caused by immune responses against various types of gram-negative anaerobic bacteria, such as *P. gingivalis, T. forsythia, T. denticola*, and *A. actinomycetemcomitans* [1]. A periodontal homeostasis imbalance in host–bacteria interactions leads to the initiation and progression of periodontal inflammation, which is regulated by many factors [2]. Pathogen infections trigger host inflammatory responses, such as the release of pro-inflammatory factors in the periodontium [3]. Chronic inflammation ultimately results in gingival recession, bone resorption, and the loss of teeth [4].

The periodontium is composed of the alveolar bone, the cementum, and the periodontal ligament (PDL), which connects the cementum to the alveolar bone [5]. The PDL contains different types of cells and serves a variety of functions. Human periodontal ligament fibroblasts (HPDLFs) constitute the majority of cells in the PDL that control alveolar bone remodeling, periodontal ligament regeneration, and protect against microbes [6]. HPDLFs can produce inflammatory mediators upon pathogenic infections, which attract immune cells including neutrophils, T lymphocytes, B lymphocytes, and macrophages that initiate inflammatory responses. Those reactions can further lead to the destruction of collagen and the resorption of alveolar bone [7].

The circadian rhythm contributes to the immune system. Circulating immune cells, pro-inflammatory cytokines, and hormones often display certain rhythms during the day [8]. Dysregulation of the circadian rhythm can disrupt the innate immune system and normal inflammatory responses [9]. Indeed, symptoms of diseases, such as rheumatoid arthritis (RA), myocardial infarction, and ischemic stroke, manifest in a circadian pattern [10,11]. The circadian rhythm is controlled by a series of transcription factors that regulate endogenous oscillations in organisms. The clock genes Dec1 and Dec2 are two members of basic helix-loop-helix (bHLH) transcription factor family and they have a 97% similarity in their bHLH region and a 52% similarity in the total proteins [12]. Dec1 and Dec2 are involved in regulating various physiological and pathological processes including the circadian rhythm, hypoxia, cell proliferation, inflammation, epithelial-to-mesenchymal transition (EMT), and carcinogenesis [13]. The expression of Dec1 and Dec2 is highly responsive to different stimuli [14,15]. Dec1 and Dec2 can be regulated by circadian rhythm proteins, such as ARNTL/CLOCK heterodimers [16,17], and they form a complex control loop in the regulation of the circadian rhythm. HIF-α directly targets the genes encoding Dec1 and Dec2 and induces the expression of Dec1 and Dec2, which is a vital process for cells adapting to hypoxia [18].

Various inflammatory pathways contribute to the pathogenesis of periodontal inflammation. Dec1 is an important mediator in CD4+ effector T cell activation [19] and is highly inducible in human B cells [20]. Dec2 is required for naïve CD4+ T cells to commit to Th2 cells [21]. Dec1 and Dec2 also participate in immune responses, exhibiting important transcription functions in the inflammation process [22,23]. In this review, we summarize recent work on the roles of Dec1 and Dec2 regarding the initialization and progression of periodontal inflammation and suggest their therapeutic potential for treating periodontal inflammation.

## 2. Autophagy

Autophagy is a lysosome-dependent degradation pathway that serves a protective role in eukaryotic cells to maintain homeostasis in response to environmental stimuli [24]. Depending on the stimulus and type of stress, autophagy can either block apoptosis or induce apoptosis [25]. Autophagy has multiple functions in cells related to cell viability, tissue damage, and the release of proinflammatory cytokines [26]. Dysregulation of autophagy has been demonstrated in the pathogenesis of inflammatory diseases including periodontal inflammation [27]. In peripheral mononuclear cells from patients with periodontal inflammation, the expression of autophagy-related protein 12 (ATG) and microtubule-associated protein 1 light chain 3 alpha (LC3) was increased. Levels of autophagy markers were also upregulated in periodontal inflammation [28] and in an experimental model of periodontal inflammation [29]. Increased autophagy in periodontal ligament stem cells (PDLSCs) was observed [30]. HPDLFs were protected from inflammation-induced apoptosis by autophagy [31]. On the other hand, in conditions of excessive stress, autophagy promotes apoptosis and inhibits cell growth [32].

Autophagy activity has been reported to appear in certain rhythm oscillations. In rat hepatocytes and cardiomyocytes, autophagy declines at feeding time and increases when at rest phase [33]. In the mouse liver, CCAAT enhancer binding protein beta (C/EBPb) is a key transcription factor that controls the diurnal autophagy rhythm [34]. Knockout of the core clock gene, circadian locomotor output cycles kaput (CLOCK), elevated the level of inflammation in mice [35].

Periodontal inflammation is characterized by a dysbiosis of oral microbiota which causes a chronic inflammatory condition [36]. Dysbiosis is characterized by the changes in the structure of the microbial species associated with diseases [37]. The augmentation of versatile polymicrobial communities were apparent in periodontal disease, whereas they were substantially decreased in healthy subjects [38,39]. Pathogens including *P. gingivalis* and its toxins, such as lipopolysaccharide (LPS), promote the release of inflammatory mediators, including TNF-α, IL-1β, and IL-6. Those cytokines cause the destruction of collagen and the resorption of alveolar bone in the progression of periodontal inflammation [7]. Treatment with LPS increased the number of T lymphocytes in gingival mononuclear cells [23], and it has been confirmed that LPS induces inflammation through autophagy [37].

### 2.1. Dec1 and Autophagy

Recent studies have demonstrated that the mammalian target of rapamycin (mTOR) is a key factor in the regulation of autophagy [38]. Autophagy can be inhibited by stimulation of the PI3K/Akt/mTOR pathway [39] or can be stimulated when the mTOR pathway is inhibited [40].

Oka et al. has recently found that *P. gingivalis* LPS upregulates autophagy and results in inflammation in HPDLFs. Accompanied by an elevation of IL-1β and other autophagy markers, levels of Dec1 were also significantly elevated. To further investigate the role of Dec1 in LPS induced autophagy, the effects of inhibiting Dec1 were characterized. A Dec1 deficiency affected cell autophagy levels both in vitro and in vivo and that effect was mediated via the Akt/mTOR pathway. A Dec1 deficiency could partly attenuate the inflammation level triggered by LPS-induced autophagy in mice. Those results showed the important role of Dec1 in regulating inflammation and autophagy [37].

### 2.2. Dec2 and Autophagy

After LPS stimulation, the ratio of LC3-II/LC3-1 was increased, indicating the upregulation of autophagy in human gingival fibroblasts [41]. The Raf/MEK/ERK signaling pathway regulates autophagy and, upon activation, LC3-II protein expression is also increased which in turn promotes autophagy [42].

Recent in vitro and in vivo studies demonstrated that Dec2 is involved in regulating LPS-induced autophagy in HPDLFs. Knockdown of Dec2 increased autophagy and IL-1β levels in HPDLFs treated with *P. gingivalis* LPS although functional bindings sites were not detected between Dec2 and IL-1β. We further determined the mechanism by which Dec2 mediates autophagy. Treatment with a Dec2 siRNA indicated that Dec2 is importantly related to activation of the ERK signaling pathway, and Dec2 also inhibited autophagy via mTOR phosphorylation and subsequently 4EBP1 activation. The suppression of autophagy by Dec2 was also confirmed in an animal periodontal inflammation model [43]. Figure 1 shows a schematic diagram of the roles of Dec1 and Dec2 in autophagy. These data provide a new perspective on the pathogenesis of periodontal inflammation.

## 3. Pyroptosis

Pyroptosis is another important pathway in periodontal inflammation [44]. Pyroptosis is a distinct type of programmed cell death mediated by inflammatory caspases (caspase-1, -4, -5 and -11) and gasdermin D (GSDMD) [45]. LPS is sensed by Toll-like receptor 4 and murine caspase-11 or human caspase 4/5 [46]. The activation of caspases cleaved GSDMD, leading to pyroptosis and NOD-like receptor family and pyrin domain-containing protein 3 (NLRP3) mediated formation of inflammasomes and the release of IL-1β [45]. Caspase-1 cleaves pro-IL-1β upon activation, and IL-1β attracts immune cells causing inflammation [47]. The dysregulation of pyroptosis causes an aberrant inflammatory response, further damages the tissue, and results in inflammatory disease [47]. NF-κB contributes important functions to the progression of pyroptosis [48]. TLR4/NF-κB signaling also triggers pyroptosis [49].

NLRP3 mRNA levels display a circadian rhythm in mouse macrophages as well as the secretion of IL-1β and IL-18 [50]. A core clock component, Nuclear Receptor Subfamily 1 Group D (NR1D1), also affects IL-1β and IL-18 mRNA levels [50]. The inhibition of NR1D1 aggravates the acute lung injury and inflammation induced by LPS via the NLRP3-dependent secretion of IL-1β [51].

### 3.1. Dec1 and Pyroptosis

A previous study demonstrated that Dec1 can upregulate IL-1β in human gingival cells via the PI3K/AKT pathway and, thus, can influence the process of inflammation [22]. Recent studies took advantage of Dec1 KO mice to gain more insight between Dec1 and pyroptosis. Caspase-1, as well as phosphorylated NF-κB, was highly activated after LPS stimulation, indicating that pyroptosis was successfully induced. The mRNA and protein levels of IL-1β were also upregulated. Interestingly, the expression of Dec1 was significantly elevated by LPS. Dec1 deficiencies in vitro and in vivo downregulated the expression of IL-1β and NF-κB, indicating reduced pyroptosis after LPS treatment [52]. Those results suggest that Dec1 has a therapeutic potential to treat periodontal inflammation as a target in inflammation pathways.

### 3.2. Dec2 and Pyroptosis

Dec2 also plays an important role in immune regulation. To explore the association between Dec2 and pyroptosis, HGFs, HPDLFs, and an experimental periodontal inflammation mouse model were studied. Cells treated with 10 μg/mL of *P. gingivalis* LPS showed a significantly increased expression of Dec2 and IL-1β. Caspase-1 and GSDMD were subsequently cleaved, and NF-κB was phosphorylated and then translocated into the nuclei. Both Dec2 siRNA treatment of cells and a Dec2 KO mouse periodontal inflammation model showed a regulatory function against the cleavage of GSDMD, demonstrating that Dec2 has a critical role in pyroptosis [53]. Figure 2 shows a schematic diagram summarizing the roles of Dec1 and Dec2 in pyroptosis.

## 4. Clock Genes and Other Inflammatory Diseases

Given the roles of a wide array of clock genes, including Dec1 and Dec2 in different physiological and pathological processes, recent studies have also identified their functions not only in periodontal inflammation but also in other diseases.

### 4.1. Brain and Muscle Arnt-like Protein-1 (BMAL1) an Alzheimer Disease (AD)

Another important aspect of clock genes is their potential involvement in neurodegenerative diseases including AD [54]. BMAL1 knockout induces age-related astrogliosis, neuronal oxidative damage caused by impaired redox-related genes dysregulation [55]. BMAL1 knockout also increases ROS levels in peripheral tissues, such as the spleen and kidney, and the redox state also shows circadian rhythms [56,57]. These data indicate that BMAL1 could participate in the pathogenesis of AD. More in-depth investigations are needed to clarify how clock genes correlate with key proteins such as amyloid-β and tau.

### 4.2. NR1D1 and Atherogenesis

Atherogenesis is a chronic inflammatory disease driven by lipid [58]. The accumulation of infiltrations and modifications of lipoproteins results in the upregulation of inflammatory cells, including macrophages, and T and B lymphocytes [59,60]. NR1D1 reduction in hematopoietic cells exacerbates atherogenesis in low-density lipoprotein (LDL) receptor-deficient mice [61]. An NR1D1 agonist suppresses atherogenesis in that animal model and that treatment induces the polarization of bone marrow-derived macrophages from M1 macrophages to M2 macrophages [62].

### 4.3. Dec1 and Cardiac Hypertrophy

Cardiac hypertrophy is a common feature of several cardiovascular diseases. Macrophages contribute to the immune regulating process of cardiac hypertrophy through the crosstalk of M1 and M2 macrophages [63]. Dec1 has been demonstrated to change the polarization of M1 and M2 macrophages in Dec1 knockout mice after transverse aortic constriction. The knockout of Dec1 alleviated the level of cardiac fibrosis, inflammation, and apoptosis, which suggests a novel mechanism for the treatment of cardiac hypertrophy [64].

### 4.4. Dec 1 and Ischemia/Reperfusion-Induced Myocardial Inflammation

Revascularization is an effective treatment for myocardial ischemia; however, revascularization also causes ischemia/reperfusion injury [65,66]. Inflammation is an important factor in myocardial ischemia/reperfusion injury, and prolonged inflammatory responses induce myocardial damage and a series of complications [67,68]. Xu et al. demonstrated that Dec1 mRNA interference significantly decreased the expression of Toll-like receptor 4 (TLR4), NF-κB and TNF-α [69]. This suggests that Dec1 plays an important role in ischemia/reperfusion-induced myocardial inflammation through the TLF4/NF-κB signaling pathway, which is related to pyroptosis.

### 4.5. Dec2 and Rheumatoid Arthritis (RA)

RA is a chronic inflammatory joint disease in which symptoms appear to be associated with an abnormal circadian rhythm. Circulating inflammatory cytokines, such as TNF-α and IL-6, show different circadian rhythms between healthy subjects and RA patients [70]. As important regulators, Dec1 and Dec2 can form a negative feedback loop and compete with major circadian rhythm genes BMAL1 and CLOCK [71]. Olkkonen et al. [72] found that Dec2 expression is abundant in synovial membranes in RA patients and Dec2 mRNA and protein levels were regulated by TNF-α and NF-κB. Dec2 overexpression increases IL-1β, which suggests an important role of Dec2 in RA.

## 5. Perspectives

Clock gene-related organ and tissue-specific cell kinetic and morphological alterations are associated with the incidence of numerous diseases including salivary dysfunction. Dec1 and Dec2 exhibited potent circadian rhythms in mouse submandibular glands [73], suggesting that these genes are of fundamental importance for elucidating the species of modifications in salivary functions. Given their roles in inflammation, understanding the intervention of Dec1 and Dec2 in salivary cytokine expression is indispensable.

Orthodontic tooth movement is a reminiscent procedure of physiological alveolar bone remodeling. The formation of new bone and osteoclastic bone resorption determines conventional tooth movement. Consequently, to provide insight on the mechanisms behind orthodontic tooth movement, it is of fundamental importance to analyze Dec1 and Dec2 signaling and its downstream targets in alveolar bone biology. A major osteogenic factor, Osteopontin (OPN), is involved in the process of bone remodeling and orthodontic tooth movement [74]. OPN has an evident circadian rhythm in vivo, suggesting a periodicity regulated orthodontic tooth movement [75]. Orthodontic treatment augments the periodontal indices and powers the aggregation and structure of the oral microbiota [76].

Certain drugs trigger gingival inflammation, resulting in gingival overgrowth [77]. Deficiency of Toll-like receptor (TLR) 4 displayed a substantial decline in accumulation of fibroblasts [78], and an increase in TLR-4 expression followed by Dec1 upregulation in impaired periodontal tissues [23] seems to indicate that a link between transcriptional regulation of Dec1 and TLR-4 is crucial to prevent gingival overgrowth. Previous studies demonstrated that the regulation of Dec1 and Dec2 in pro-inflammatory cytokines mediated periodontal inflammation [22,23,40,46,55,56] and established the basis for chronobiological scheme-aided drug design in clinically effective personalized oral care.

Clinical paradigms related to periodontal health and regeneration are intricately connected to the balance of apoptotic pathways. Any significant shift towards the catabolic mechanisms often results in irreparable damage to the periodontium. The recent insights into the Dec1- and Dec2-associated pathways has provided significant clarity in the function of these clock genes in periodontium associated apoptosis, inflammation, and immune modulations. Nevertheless, their precise mechanism of action remains largely basic research findings, their clinical implications are enormous. Hence, identifying the appropriate markers is crucial in assessment of risk for periodontal disease or external root resorption. Likewise, an insightful understanding of these pathways will be an asset in developing individualized treatment protocols.

## 6. Conclusions

Recent studies have demonstrated that autophagy and pyroptosis contribute significantly to the pathogenesis of periodontal inflammation. The roles of Dec1 and Dec2 in autophagy and pyroptosis highlight their potential for future studies of periodontal inflammation.

Understanding the mechanism of the pathological process of periodontal inflammation would not only provide insights into potential therapies for periodontal inflammation but also for the treatment of other diseases. Studies representing functions of Dec1 and Dec2 in periodontal pathological processes are listed in Table 1. Dec1 and Dec2 have important roles in diseases such as cancer, cardiac hypertrophy, rheumatoid arthritis, and periodontal inflammation [64,72]. Accumulating evidence has indicated associations between periodontal disease and systemic diseases, rheumatoid arthritis, obesity, metabolic syndrome, and cancer [79,80,81]. Systematic studies of Dec1 and Dec2 will provide promising directions for the future and could provide more insights into understanding the pathogenesis of diseases and identifying new drug targets. With the role of Dec1 and Dec2 in the circadian rhythm, time point-based therapeutic remedy would improve the efficacy of treatments. 

## Figures and Tables

**Figure 1 ijms-22-10349-f001:**
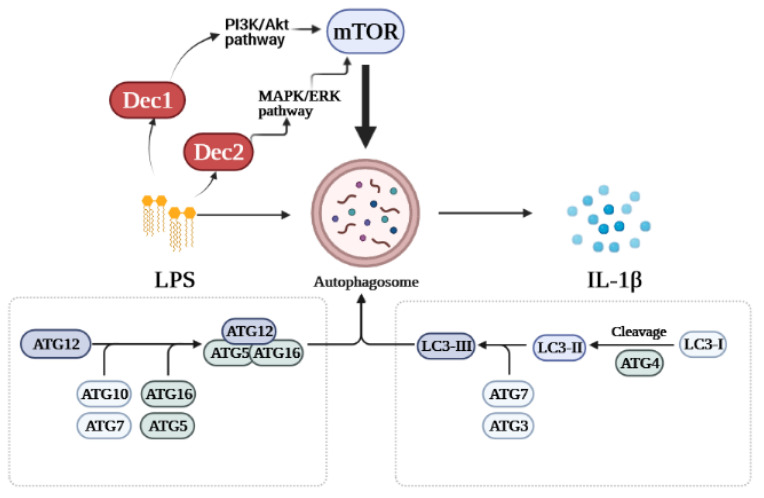
Schematic diagram of the roles of Dec1 and Dec2 in autophagy. Treatment with LPS stimulates the expression of Dec1 and Dec2. Dec1 mediates autophagy via the PI3K/Akt/mTOR pathway, while Dec2 regulates autophagy through the MAPK/ERK/mTOR pathway.

**Figure 2 ijms-22-10349-f002:**
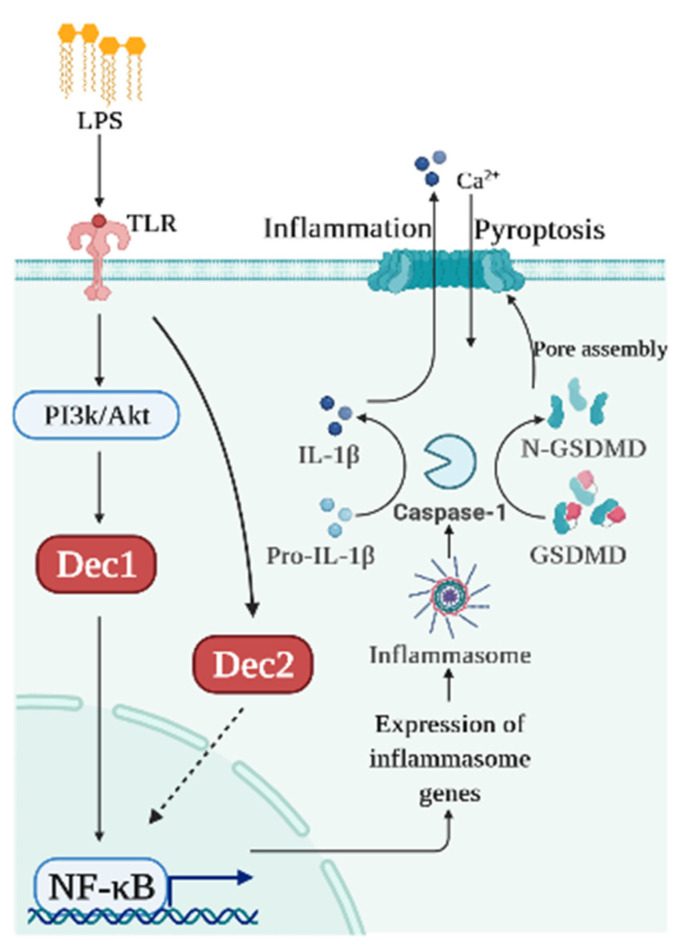
Schematic diagram of the roles of Dec1 and Dec2 in pyroptosis. Bacterial products like LPS induce Dec1 expression and Dec1 upregulates IL-1β and NF-κB. Dec2 attenuates the phosphorylation of NF-κB which, in turn, deactivates caspase-1 and GSDMD.

**Table 1 ijms-22-10349-t001:** Studies represent functions of Dec1 and Dec2 in periodontal pathological processes.

Title	Authors
Loss of Dec1 prevents autophagy in inflamed periodontal ligament fibroblast	Oka et al. [40]
Inhibition of Dec1 provides biological insights into periodontal pyroptosis	Oka et al. [55]
Transcription factor DEC1 is required for maximal experimentally induced periodontal inflammation	Zhang et al. [23]
The role of the hypoxia responsive gene DEC1 in periodontal inflammation	Kim et al. [82]
Differentiated embryonic chondrocytes 1 expression of periodontal ligament tissue and gingival tissue in the patients with chronic periodontitis	Hu et al. [83]
IL-1β-mediated up-regulation of DEC1 in human gingiva cells via the Akt pathway	Bhawal et al. [22]
Dec2 inhibits macrophage pyroptosis to promote periodontal homeostasis	He et al. [84]
Dec2 attenuates autophagy in inflamed periodontal tissues	Oka et al. [46]
A deficiency of Dec2 triggers periodontal inflammation and pyroptosis	Oka et al. [56]

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
