# Peer review of "The Potential Roles of Dec1 and Dec2 in Periodontal Inflammation"

_ijms, 2021, doi:10.3390/ijms221910349_

Round 1

Reviewer 1 Report

The review paper authored by Wang et al. approaches a very interesting topic regarding potential genetic mediators of periodontal inflammation.

I hope that my remarks and suggestions will help the authors to increase the quality of their manuscript.

1) Line 88 - Authors should elaborate more on dysbiosis. Please explain more detailedly about this important topic.

2) Line 147- Can Dec1 can upregulate salivary citokynes as well? Did you find any studies related to salivary research?

3) Is their any particular potential role of Dec1 and Dec2 in periodontal inflammation during orthodontic treatment?

4) Is there any relationship between Dec1 and Dec2 and Gingival Overgrowth? 

5) Please elaborate more on the perspective part and emphasise the future practical aspects related to clinical protocols of identifying Dec1 and Dec2

6) Please separate and rephrase the Conclusions part into a new chapter in order to increase the impact of the paper to the reader.

7) Please add a synthetic table with other findings from the literature regarding the role of Dec1 and Dec2 in periodontal pathological processes.

Please receive my best regards!

Reviewer 2 Report

Dear authors,

This is an interesting review summarizing the last data concerning Dec 1 and Dec 2. The manuscript is well structured and the figure are very important and well design for the comprehension. Just some remarks:

Abstract

  1. L19 Delete « conducted by our group and by other groups”

Main manuscript

  1. Even if it is right, avoid using “our research group” all along the manuscript. This sentence decreases the quality of your review. This gives the impression that you have done a summary of your research and not a real review

Conclusion

  1. L229 The reference 73 is very old. Replace with more recent references (PMID: 30384973, PMID: 28303212, PMID: 29903685, PMID: 30987702, PMID: 31600905…)

Round 2

Reviewer 1 Report

Dear authors,

Thank you very much for the revised version.

I consider that it looks much better now.

Best of luck in your future projects!